# Co-Players in Chronic Pain: Neuroinflammation and the Tryptophan-Kynurenine Metabolic Pathway

**DOI:** 10.3390/biomedicines9080897

**Published:** 2021-07-26

**Authors:** Masaru Tanaka, Nóra Török, Fanni Tóth, Ágnes Szabó, László Vécsei

**Affiliations:** 1MTA-SZTE, Neuroscience Research Group, Semmelweis u. 6, H-6725 Szeged, Hungary; tanaka.masaru.1@med.u-szeged.hu (M.T.); toroknora85@gmail.com (N.T.); toth.fanni@med.u-szeged.hu (F.T.); 2Interdisciplinary Excellence Centre, Department of Neurology, Faculty of Medicine, University of Szeged, H-6725 Szeged, Hungary; szabo.agnes.4@med.u-szeged.hu

**Keywords:** chronic pain, nociceptive pain, neuropathic pain, nociplastic pain, psychogenic pain, neuroinflammation, kynurenine

## Abstract

Chronic pain is an unpleasant sensory and emotional experience that persists or recurs more than three months and may extend beyond the expected time of healing. Recently, nociplastic pain has been introduced as a descriptor of the mechanism of pain, which is due to the disturbance of neural processing without actual or potential tissue damage, appearing to replace a concept of psychogenic pain. An interdisciplinary task force of the International Association for the Study of Pain (IASP) compiled a systematic classification of clinical conditions associated with chronic pain, which was published in 2018 and will officially come into effect in 2022 in the 11th revision of the International Statistical Classification of Diseases and Related Health Problems (ICD-11) by the World Health Organization. ICD-11 offers the option for recording the presence of psychological or social factors in chronic pain; however, cognitive, emotional, and social dimensions in the pathogenesis of chronic pain are missing. Earlier pain disorder was defined as a condition with chronic pain associated with psychological factors, but it was replaced with somatic symptom disorder with predominant pain in the Diagnostic and Statistical Manual of Mental Disorders, 5th Edition (DSM-5) in 2013. Recently clinical nosology is trending toward highlighting neurological pathology of chronic pain, discounting psychological or social factors in the pathogenesis of pain. This review article discusses components of the pain pathway, the component-based mechanisms of pain, central and peripheral sensitization, roles of chronic inflammation, and the involvement of tryptophan-kynurenine pathway metabolites, exploring the participation of psychosocial and behavioral factors in central sensitization of diseases progressing into the development of chronic pain, comorbid diseases that commonly present a symptom of chronic pain, and psychiatric disorders that manifest chronic pain without obvious actual or potential tissue damage.

## 1. Introduction

Chronic pain is an unpleasant sensory and emotional experience that persists or recurs more than three months and may extend beyond the expected time of healing [1,2]. Chronic pain occurs as a part of symptoms due to an underlying medical condition or remains despite successful treatment of the condition that originally caused it [3]. Chronic pain frequently becomes the sole or predominant clinical complaint [4]. The prevalence of chronic pain estimates as much as 20%, and the incidence reaches about 10% every year of the world adult population [5]. Nearly 10% of individuals with chronic pain was found to suffer from moderate to severe debilitating pain [6]. Furthermore, individuals with severe chronic pain are twice more likely to die of respiratory disease or heart disease than those with mild pain or without pain [5]. The Global Burden of Disease Research ranked low back pain and migraine first and second place of Years Lived with Disability (YLD), respectively, and thus, chronic pain imposes a substantial socioeconomic burden directly and indirectly on society [7].

The International Classification of Diseases, Eleventh Revision (ICD-11), classifies chronic pain into primary and secondary. Primary chronic pain is fibromyalgia or low-back pain; the secondary chronic pain occurs secondary to an underlying medical condition subcategorizing into cancer-related, post-trauma, neuropathic, headache and orofacial, visceral, and musculoskeletal pain. ICD-11 offers minimal options for recording psychological or social factors in chronic pain [8]. Meanwhile, the Diagnostic and Statistical Manual of Mental Disorders, 5th Edition (DSM-5) recognizes chronic pain in the diagnosis of somatic symptom disorder (SSD), having replaced pain disorder, a condition with chronic pain due to psychological factors [9]. SSD is caused by somatosensory amplification, which is associated with fibromyalgia [10]. The trend toward a neurological explanation obviously discounts cognitive, emotional, and social dimensions in the pathomechanism of chronic pain. Hyperalgesia is a condition of abnormally increased sensitivity to pain caused by injury to tissues or nerves. Nociceptive sensation is also caused by exposure to opioids used for pain treatment, which paradoxically makes individuals more sensitive to certain stimuli. Hyperalgesia is a challenging issue for pain specialists who treat patients at terminal care [11]. Chronic pain is often elicited by stimuli that previously did not provoke discomfort sensation. It is called allodynia. Allodynia is commonly observed in patients with neuropathies, fibromyalgia, migraine, complex regional pain syndrome, and postherpetic neuralgia [12]. Chronic pain may proceed to clinical conditions accompanied often by mood alterations, such as depression, anxiety, anger, cognitive disturbance including memory impairment, sleep disturbances, fatigue, loss of libido, and/or disability, called chronic pain syndrome (CPS). CPS appears to be linked to the dysfunction of the hypothalamic–pituitary–adrenal axis and the central nervous system (CNS), but exact mechanisms remain unknown [13]. 

Neuroinflammation has been intricately linked to the pathogenesis of chronic pain. Chronic pain was proposed to be caused by the disturbance of peripheral nociception, neuropathy in the somatosensory system, motor system, central and peripheral nociplasticity, and/or psychosocial system [14]. Increasing evidence suggests that chronic inflammation is strongly tied to aberration in each mechanism of chronic pain. Furthermore, the tryptophan (TRP)–kynurenine (KYN) pathway and its metabolites were observed to play an important role in neuroinflammation and chronic pain [15]. This review article presents the components of the pain pathway; mechanisms of chronic pain based on the components; the development of chronic pain through peripheral and central sensitization; evidence of the presence of chronic neuroinflammation in each pain mechanism; the involvement of the TRP–KYN metabolic pathway; and the need of a psychogenic component in the pathogenesis of chronic pain.

## 2. The Pain Pathway, Mechanisms, Neuroinflammation, and Tryptophan Metabolism

Pain perception is signaling through the pain pathway, whose components consist of transduction, conduction, transmission, modulation, and perception. Transduction is the process by which noxious or potentially damaging stimuli activate the nociceptors to convert to neural signals. Transmission refers to the signal transfer from the peripheral neurons to the second-order neurons in the spinal cord, which wire the signals to the thalamus and brain stem in the brain. Pain modulation takes place by inhibition of pain signaling in the spinal cord and the activation of the descending inhibitory fibers. The third-order neurons project to the somatosensory cortex, enabling the perception of pain. Perception is the subjective awareness in connection with arousal, physiological, and behavioral brain centers, involving the integration of psychological processes such as attention, expectation, and interpretation [16,17,18] (Figure 1).

Pain is a complex and intricate process attributable to nociceptic, neuropathic, and/or neuroplastic mechanisms. The most common type of pain is nociceptive pain caused by damage or potentially harmful to peripheral tissues involving nociceptors responsible for transduction. Neuropathic pain is caused by lesions or diseases affecting the somatosensory nervous system responsible for the transmission of peripheral to central pain signals. Nociplastic pain refers to the condition caused by altered nociceptive processing without actual or potentially harmful tissue damage activating peripheral nociceptors (nociceptive pain) or without lesions or diseases of the somatosensory nervous system (neuropathic pain). Cortical perception is one of the main components in the pain pathway; however, the ICD-11 excludes psychogenic pain [19] (Figure 2). Thus, participation of cortical perception in chronic pain mechanisms remains ambiguous.

Inflammation is generally involved in the pathogenesis of various diseases and plays a key role in diseases that cause chronic pain [20]. Resident and recruited immune cells release inflammatory mediators at peripheral nerve innervating damaged or inflammatory tissue to trigger action potentials in sensor neurons or sensitize neurons by increasing transduction and excitability. Inflammatory mediators also act directly on peripheral nerves to damage peripheral transmission [21]. Immune cells infiltrate the spinal cord and the dorsal root ganglia to damage the central transmission and/or modulate pain sensitivity [22]. Activated immune cells release inflammatory cytokines, chemokines, and other factors that influence cognition, mood, and behaviors through immune-to-CNS signaling [16,23]. Accumulating evidence suggests that chronic dysregulation of the immune response is involved in the pathogenesis of psychiatric disorders such as mood disorders, substance abuse disorders, psychotic disorders, attention-deficit disorders, and autism spectrum disorders [24,25,26] (Figure 2).

Inflammation is invariably linked to the activation of TRP metabolism [27,28]. The essential amino acid TRP is a precursor to serotonin, melatonin, and nicotinamide adenine dinucleotide (NAD^+^), among others. More than 95% of TRP is metabolized through the TRP–KYN pathway, synthesizing various bioactive metabolites such as neuroprotective antioxidants and neuroprotectants, toxic oxidants and neurotoxins, as well as immunomodulators. The disturbance of KYN metabolites has been linked to immune disorders, cancers, neurodegenerative diseases, and psychiatric disorders [29]. Furthermore, TRP–KYN metabolites are under extensive research in search of peripheral biomarkers as well as novel drug prototypes for a wide range of diseases [30,31,32,33,34,35,36]. Inflammation activates the TRP–KYN pathway, elevating the levels of oxidative compounds or neurotoxic ligands to receptors of the excitatory glutamatergic nervous system, which damage the peripheral nervous system or CNS through the broken blood–nerve or blood–brain barrier, respectively [16]. Furthermore, immunomodulators are known to trigger the shift of acute inflammatory status toward tolerogenic and chronic inflammation, perpetuating low-grade inflammation [28,37]. KYN is synthesized from TRP by the tryptophan 2,3-dioxygenase (TDO) in the liver and the indoleamine 2,3-dioxygenases (IDOs) in the brain and the immune system, which are induced by cortisol, and interferon (IFN)-α, IFN-γ, and tumor necrosis factor (TNF)-α, respectively [38]. Anthranilic acid (AA), 3-hydroxykynurenine (3-HK), or kynurenic acid (KYNA) are produced from KYN by the kynureninase (KYNU), the KYN-3-monooxygenase (KMO), or the kynurenine aminotransferases (KATs), respectively. The KATs also convert 3-HK to xanthurenic acid (XA). XA converts into cinnabarinic acid by autoxidation. AA and 3-HK convert into 3-hydroxyanthranilic acid (3-HAA) and then into picolinic acid and quinolinic acid (QA). QA converts into NADH, which is a feedback inhibitor of TDO [31] (Figure 3). Generally, 3-HK and QA are described as neurotoxic, while KYNA is considered to be neuroprotective. The 3-HK/KYNA ratio is often applied as an indicator of neurotoxicity. However, emerging evidence suggests that some metabolites of the TRP–KYN pathway possess Janus-face properties, depending on the dose or the situation. For example, KYNA is excitatory in lower concentrations but inhibitory in higher concentrations at α-amino-3-hydroxy-5-methyl-4-isoxazole propionic acid (AMPA) receptors. 3-HK is known to be an oxidant but observed to be an antioxidant in certain conditions [27,39].

The stress hormone cortisol, the strong immune activator lipopolysaccharide, proinflammatory cytokines, positive feedback loops, diminished levels of antioxidant system enzyme superoxide dismutase, and anti-inflammatory cytokines all lead to the potentiation of the TRP–KYN pathway [28]. Furthermore, the action of the KYN enzymes and metabolites are complicated by the interactions with adjacent biosystems such as the oxidative stress complex, the antioxidant enzyme systems, the serotonin neurotransmission, the glutamate neurotransmission, the tetrahydrobiopterin pathway, the cannabinoid system, and the aryl hydrocarbon receptor signaling [28,39].

## 3. Transduction and Nociceptive Pain

The transduction of the pain sensation takes place when noxious stimuli depolarize the afferent terminal of nociceptive myelinated A-beta (Aß) and A-delta (Aδ) fibers and unmyelinated C fibers through the terminal membrane proteins and voltage-gated ion channels converting them into electric signals in the neurons (Figure 1a).

Nociceptive pain is the most common pain that originates from a tissue injury or inflammation in which the nociceptor of peripheral sensory nerves detects noxious or potentially harmful stimuli [40]. In chronic pain, the peripheral nociceptors continue to transmit painful stimuli even after the original injury has healed [41]. Osteoarthritis is a classical nociceptive pain condition when abnormal loading of a damaged joint opens mechanogated ion channels on nociceptive nerve endings [42]. Overextending or tearing a ligament sensitizes nociceptors, which causes acute nociceptive pain, such as in the case of an ankle sprain. In addition to mechanical irritation or physical injury, the primary cells of the epidermis, keratinocytes, induce pain by releasing endogenous mediators, such as adenosine triphosphate (ATP), Interleukin (IL)-1 beta (β), prostaglandin E2, endothelin, and nerve growth factor. However, keratinocytes act in a dual matter in pain sensation: They release β–endorphin that help pleasurable feeling during modest sun-bathing but activate transient receptor potential cation channel subfamily V member 4 (TRPV4) and release pro-inflammatory cytokines, eliciting the pain sensation of a sunburn [43,44] (Figure 2a and Table 1).

Inflammation also activates nociceptors in the nerve endings. Inflammatory mediators bind to their receptors on nociceptive sensory neurons in the peripheral nervous system, resulting in pain [20]. Pro-inflammatory factors including TNF-γ and IL-1β secreted by monocytes and macrophages at the site of a peripheral injury facilitate pain transduction and conduction by modifying ion channels including transient receptor potential cation channel subfamily A member 1 (TRPA1), transient receptor potential cation channel subfamily V member 1 (TRPV1), and Nav1.7–1.9. However, those cells secret anti-inflammatory factors such as IL-10 and/or pro-resolution mediators, including resolvins, protectins, and maresins, to reduce nociception in the resolution phase of acute inflammation. Different phenotypes of macrophages, such as pro-inflammatory M1 and anti-inflammatory M2, contribute to the induction and resolution of pain, respectively [45]. Schwann cells of the peripheral nervous system also secret TNF-γ and IL-1β to sensitize nociceptors at axons in neuronal injury. Activated Schwann cells secrete matrix metalloprotease (MMP) 9 that help open the blood–nerve barrier, resulting in the recruitment of immune cells that release inflammatory cytokines [22,46]. Furthermore, nociceptive afferent sensory neurons directly modulate inflammation by releasing inflammatory mediators, such as substance P, calcitonin gene-related peptide (CGRP), neurokinin A, and endothelin-3. The process is called neurogenic inflammation (Figure 2d).

The disturbance of TRP metabolism is observed in neurogenic inflammation. The increased levels of the stress hormone cortisol and inflammatory cytokines such as IFN-α, IFN-γ, and TNF-α activate the TRP–KYN pathway producing higher levels of oxidant KYN metabolites which leak into the peripheral nervous system through the damaged gap junction following the immune reaction. The oxidative KYN metabolites 3-HK, 3-HA, and QA are harmful compounds to nerve endings of the afferent sensory neurons (Figure 3).

## 4. Conduction, Transmission, and Neuropathic Pain

In conduction, the electrical signals are conducted from the peripheral neurons to the central neurons where a network of interneurons facilitates or inhibits transmission to the second-order neurons in the dorsal horn [47]. The presynaptic terminals of C fibers release glutamate, substance P, and CGRP, which activate postsynaptic AMPA receptors, NK1 receptors, and CGRP receptors, respectively [48] (Figure 1b). In transmission, the activation of the postsynaptic receptors generates an action potential of the second-order neurons and interneurons, which relay signals through the contralateral spinothalamic tract to the thalamus; or the spinoreticular and spinomesencephalic tracts to the medulla and brain stem; or the spinohypothalamic tract to the hypothalamus [49] (Figure 1c).

Neuropathic pain originates from lesions or diseases of the somatosensory nervous system made up of peripheral and central components. Peripheral neuropathic pain is commonly caused by diabetic neuropathy, metabolic disorders, shingles, HIV-related distal symmetrical neuropathies, nutritional deficiencies, toxins such as arsenic and thallium, a paraneoplastic manifestation of cancer, immune-mediated inflammatory diseases such as Guillain-Barre syndrome, amyloidosis, Fabry’s disease, and nerve trunk injuries [50]. Presumably, burning and poorly localized pain is transmitted by C fibers, while sharp and lancinating pain is relayed by Aδ fibers [51]. Diabetic neuropathy is the most common neuropathy associated with severe pain, which presents a distal symmetrical polyneuropathy with numbness and loss of sensation in the distal extremities, often accompanied by peripheral vascular diseases, leading to infection and ultimately amputation [52]. Neuropathic pain is also caused by direct invasion to peripheral nerves by tumor, side effects of chemotherapy, radiation injury, or surgery [53] (Figure 2b and Table 1). Central neuropathic pain is a common sequela to injury to the CNS such as vascular accidents, including ischemic and hemorrhagic stroke, infections, including abscess, encephalitis, and myelitis, demyelinating diseases, including multiple sclerosis, tumors, and brain or spinal cord [54,55,56] (Figure 2b and Table 1). Mixed pain is a term never formally defined, but it indicates pain caused by a combination of nociceptive and neuropathic mechanisms observed in patients who suffer from osteoarthritis, sciatica, and cancer. 

Neuropathic pain is often manifested as a part of the symptoms of psychological disorders. The lifetime and current prevalence of psychiatric disorders in patients with chronic peripheral pain were 39% and 20%, respectively [57]. Diseases that cause neuropathic pain include diabetes, herpes zoster infection, nerve compression, nerve trauma, channelopathies, and autoimmune diseases. The most common psychiatric disorders were generalized anxiety disorders and mood disorders [57]. Furthermore, antidepressants showed efficacy for neuropathic pain in patients with depression, suggesting neuropathic pain and depression have a bidirectional relationship [58]. Individuals with chronic neuropathic pain were associated with substance abuse or suicide ideation [59] (Figure 2b and Table 1).

Inflammation plays an important role in neuropathic pain. Around afferent peripheral nerves, monocytes and macrophages release pro-inflammatory factors, including TNF-γ and IL-1β, while they secrete anti-inflammatory factor IL-10 and pro-resolving lipid mediators at the resolution of acute inflammation [60]. T lymphocytes (T-cells) play an important role in neuropathic pain. T-cells secret a pro-inflammatory cytokine IL-17 and accumulate in the dorsal root ganglion (DRG) to release pro-analgesic leukocyte elastase, inducing mechanical allodynia. In the resolution phase, T-cells secrete anti-inflammatory cytokines IL-4 and IL-10. In response to noxious stimuli, the satellite glial cells (SGCs) are activated and proliferated at DRG to release pro-inflammatory cytokines TNF and IL-1β and a nociceptive neurotransmitter ATP signaling through P2 receptors [61]. SGCs also release MMPs that open the blood–nerve barriers, allowing entry of immune cells [62]. Bone marrow stem cells trigger analgesic actions by secreting anti-inflammatory cytokine-transforming growth factor-beta 1 by suppressing glial activation induced by nerve injury and migrating to DRG via a (C-X-C motif) chemokine ligand (CXCL) 12 chemotactic signal after intrathecal injection [63]. 

Spinal cord microglia play major roles in pathological pain. Following peripheral injury, ATP, colony-stimulating factor 1, chemokines including (C-C motif) chemokine ligand (CCL) 2 and fractalkine **(**CX3CL1), and proteases activate spinal microglia [64]. Meanwhile, the expression of the receptors for ATP and CX3CL1 increases, converging an intracellular signaling cascade, leading to the phosphorylation of p38 mitogen-activated protein (MAP) kinase, which, in turn, elevates production and release of TNF-γ, IL-1β, IL-18, brain-derived growth factor (BDNF), and prostaglandin E2. TNF-γ and Il-1β increase synaptic transmission and decrease inhibitory synaptic transmission of lamina II spinal cord neurons [22]. BDNF suppressed gamma-aminobutyric acid inhibitory synaptic transmission in projection to lamina I spinal cord neurons. Microglia release anti-inflammatory cytokine IL-10 in the resolution phase of inflammation [65].

An astrocyte is in contact with more than one million synapses, and thus, chronic pain in astrocyte activation is more persistent [66]. Astrocytes communicate with neurons through gap junction mediated by connexin-43 (Cx43). Cx43 is upregulated in astrocytes after nerve injury, serving as a paracrine modulator. The paracrine modulation results in elevating the release of glutamate, ATP, MMP2 and chemokines, including CCL2 and CXCL1. The chemokines function as neuromodulators that potentiate excitatory synaptic transmission. Meanwhile, following nerve injury, spinal cord neurons upregulate CXCL13 that activates astrocytes via C-C chemokine receptor type 5 to sustain neuropathic pain [67]. The spinal cord and cortical astrocytes upregulate thrombospondin 4 that leads to new synapsis formation and subsequent somatosensory cortical circuit rewiring, causing neuropathic pain [68]. Astrocytes cause neuronal hyperexcitability resulting from disturbance of homeostasis of extracellular potassium and glutamate. IFN-α produced by astrocytes inhibits nociceptive transmission in the spinal cord [66].

Oligodendrocytes form myelin sheath insulating axons in the CNS [69]. Little is known about their roles in pain. IL-33 produced from oligodendrocytes contributes to pain sensitivity via MAP kinases and nuclear factor kappa-light-chain-enhancer of activated B cells in chronic constriction injury model of nerve injury-induced neuropathic pain [70]. Diphtheria toxin ablation of oligodendrocytes leads to neuropathic pain, suggesting analgesic roles of the cells. Following nerve injury, T-cells infiltrate the spinal cord, contributing to the development of mechanical sensitivity. T-cells release pro-inflammatory cytokine TNF-γ, they secrete anti-inflammatory cytokines IL-4 and IL-10 in the resolution phase of inflammation [71]. Following chemotherapy, intrathecal injection of cytotoxic T-cells enhanced neuropathic pain, while the injection of regulatory T-cells diminished neuropathic pain [72] (Figure 2d).

The involvement of the TRP–KYN pathway was reported in inflammation-induced neuropathic pain. The enzyme activities of the TRP–KYN pathway were studied in a lipopolysaccharide-stimulated chronic constriction injury at the spinal cord and DRG levels of rats. The intrathecal administration of L-KYN and the intraperitoneal injection of L-KYN and an organic anion transport inhibitor probenecid significantly reversed tactile allodynia in L5-L6 spinal nerve root-ligated rats, suggesting that the N-methyl-D-aspartate (NMDA) receptor, an organic anion transport inhibitor agonist KYNA, mediates relieving the allodynia [73]. The increased ratio of QA/KYN and the mRNA expression of KMO, KYNU, and 3-hydroxyanthranilate dioxygenase (HAOO) was elevated in neuronal nuclear antigen-positive neurons of the contralateral hippocampal dentate gyrus in a neuropathic mouse model [74]. TDOIDO1 and 2, KMO, KYNU, and HAOO were found to be derived from cerebral microglial cells, and mRNA expression of IDO2, KMO, and HAOO were upregulated at the spinal cord after one week. Microglia inhibitor, minocycline, decreased the levels of IDO2 and KMO enzymes and tactile and thermal hypersensitivity; furthermore, IDO2 inhibitor 1-methyl-d-tryptophan and KMO inhibitor UPF 648 significantly decreased mechanical and thermal hypersensitivity [75]. This suggests the participation of IDO2 and KMO enzymes in the pathogenesis of neuropathic pain. The intracerebroventricular administration of KMO inhibitor Ro 61-8048 alleviated spared nerve injury-induced depressive-like behavior, and the intrathecal injection of Ro 61-8048 attenuated both the depressive-like behavior and mechanical allodynia in rats [76]. The NMDA receptor seems to play a major role in neuropathic pain and in the development of opioid tolerance. Dextromethorphan is an NMDA antagonist at high doses. Both animal and human studies showed that NMDA antagonist ketamine was beneficial for analgesics [77] (Figure 3).

## 5. Modulation and Nociplastic Pain

Modulation of pain transmission occurs at all levels of the pain pathway from peripheral to the brain, as well as from upward-to-downward pain regulations, involving both excitatory and inhibitory mechanisms that facilitate or suppress the responses of second-order neurons, respectively [48]. Peripheral pain modulation is achieved through local growth factor, hormonal, and peptide release, which alters signaling through neurotransmitter, ion, or receptor-based mechanisms. The pain modulation takes place neuronal signaling through corticospinal, corticoperiheral, and intraspinal pathways and neuroplasticity regulation [78] (Figure 1d and Figure 4).

Nociplastic pain is defined as pain that arises from altered nociception despite no clear evidence of actual or threatened tissue damage causing the activation of peripheral nociceptors or no clear evidence of diseased lesions of the somatosensory nervous system causing the pain [79,80]. Nociplastic pain is generally chronic and widespread and is caused by the disturbance of central pain processing mechanisms, such as elevated excitability of ascending and descending pain facilitatory pathways and/or reduced inhibition of the descending anti-nociceptive pathway [14,81,82] (Figure 4). The condition refers to central sensitization in which pain is elicited by innocuous stimuli or different kinds of stimuli, resulting in central hyperalgesia or allodynia, respectively [83]. The process involves increased activity of the insula, anterior cingulate cortex, and the prefrontal cortex, which becomes active during acute pain sensation as well as of the brain stem nuclei, dorsolateral frontal cortex, and parietal cortex, which do not participate during acute pain sensation [84] (Figure 4). Fatigue, negative affect, unrefreshing sleep, and cognitive dysfunction are common accompanying findings in centralized nociplastic pain [85]. This typical pattern of nociplastic pain is observed in fibromyalgia, a medical condition of unknown cause but known to be involved in genetic and environmental factors [86]. Temporomandibular disorder and nonspecific back pain are also characterized by central sensitization (Figure 2c and Table 1).

The inflammatory response is remarkable in nociplastic pain. The levels of proinflammatory cytokines including IL-6 and IL-8 were observed to be higher, while anti-inflammatory cytokines IL-1 receptor antagonist was higher and IL-4 was lower in patients with fibromyalgia. Several chemokine levels were elevated in fibromyalgia patients. They were monocyte recruiting such as protein eotaxin (CCL11), TARC (CCL17), and MDC (CCL22) and neutrophil chemoattractant MIG (CXCL9) and I-TAC (CXCL11) [87]. Furthermore, the disruption of the proinflammatory and anti-inflammatory cytokine network was considered to play a key role in the pathogenesis of central sensitization in fibromyalgia. Chronic inflammation has been considered to induce central pain in rheumatoid arthritis [88]. Thus, inflammation certainly contributes to the development of nociplastic pain, as in fibromyalgia (Figure 2d).

The alteration of TRP metabolism has been linked to nociplastic pain such as the temporomandibular disorders myalgia and fibromyalgia. The levels of TRP were observed to be significantly lower in the plasma of fibromyalgia patients compared to control, and the KYN/TRP ratio was negatively correlated with anxiety levels. The plasma TRP levels were negatively correlated with the wrist pain intensity, whereas the KYN/TRP ratio was positively correlated with the average and wrist pain intensity in temporomandibular disorders [89]. TRP depletion appears to be involved in the pathogenesis of fibromyalgia and temporomandibular disorders; however, little is known about the roles of NMDA receptor agonists 3-HK and QA and NMDA receptor antagonist KYNA. Furthermore, the direct link between KYNs and nociplastic pain has not been reported (Figure 3).

## 6. Cortical Perception and Psychogenic Pain

The perception of pain is processed in the brain and the spinal cord. The thalamus, sensorimotor cortex, insular cortex, and anterior cingulate decode signals of unpleasant sensation carried through ascending spinothalamic tract, whereas the amygdala and hypothalamus decode signals of urgency and intensity brought through ascending the spinobulbar tract. The third-order neurons transfer signals and communicate with the cortex centers. Overall, the integration of sensations, emotions, and cognition in the brain lead to the perception of pain [90] (Figure 1e). Psychogenic pain is pain without relevant anatomic tissue injury or inconsistent with functional causes in distribution and is considered to be caused by psychological factors such as depression, anxiety, and emotion [91]. Depression, anxiety, and cognitive disturbance are common symptoms that manifest in a wide range of diseases and comorbidity [92]. Individuals with depression and anxiety often experience psychogenic pain all over their bodies without any relevant physical cause [93,94]. Other psychiatric disorders frequently observed in individuals with chronic pain include substance abuse, somatoform disorder, and panic disorders [95]. Furthermore, chronic pain is associated with the disturbance of cognitive functions such as attention, working memory, reasoning ability, information processing, and verbal communication [96,97]. 

Inflammation is obviously involved in psychiatric disorders such as depression and anxiety. Meta-analyses reported strong evidence of significantly increased levels of c-reactive protein (CRP), IL-1, IL-6, TNF-α and soluble IL-2 receptor in the serum of major depressive disorder (MDD) patients [98,99,100,101,102]. A higher concentration of CCL2/MCP-1 was also reported in patients with MDD. CRP levels in blood, serum or plasma samples was significantly raised in generalized anxiety disorder (GAD) patients by meta-analysis, and IFN-γ and TNF-α levels were significantly increased in at least two or more studies [103]. Lower levels of IL-10 and higher ratios of TNF-α/IL-10, TNF-α/IL-4, IFN-γ/IL-10, and IFN-γ/IL-4 were observed in the serum of GAD patients, showing significantly increased pro- to anti-inflammatory cytokine ratios, which suggests a distinct cytokine imbalance [104] (Figure 2d).

Similarly, activation of the TRP–KYN pathway has been reported in depression and anxiety. Meta-analyses reported decreased TRP levels in plasma and decreased levels of KYN and KYNA in MDD patients, while antidepressant-free patients showed an increased level of QA. The postmortem brain tissues from patients with MDD showed the increased QA immunoreactivity in the prefrontal cortex and hippocampus [105,106]. Magnetic resonance spectroscopy suggested a higher turnover of cells with KYN and the 3-HAA/KYN ratio in adolescent depression. Those findings are in accordance with the activation of the TRP-KYN pathway toward 3-HK and QA branches by pro-inflammatory cytokines activating IDOs, and KMO, resulting in higher neurotoxic 3-HK and QA levels [107]. Decreased plasma KYN levels were observed in endogenous anxiety and normalized after treatment [108]. The alteration of the TRP-KYN pathway by stress or inflammation may cause serotonin and melatonin deficiency, making an individual more susceptible to anxiety (Figure 3).

## 7. Conclusions and Future Perspective

The pain pathway, pain mechanisms, inflammation, KYN metabolites and enzymes of the TRP–KYN pathway, and diseases associated with chronic pain are overviewed in this review article. Pain sensation can be attributed to damage and/or potential harm in various components of the pain pathway and corresponding pain mechanisms, involving inflammation and alteration of the TRP–KYN pathway [109]. Chronic inflammation triggers not only nociceptive pain but induces other pain mechanisms, including psychogenic pain. Thus, a search for unique inflammatory signatures and various interventional targets in chronic inflammation is currently under extensive research [110,111,112,113]. Intervention through the TRP–KYN pathway is under comprehensive research to alleviate oxidative stress and excitotoxicity in various illnesses [114,115,116,117,118,119,120,121]. Meanwhile, the effectiveness of motor cortex stimulation and spinal cord stimulation to alleviate chronic pain caused by various underlying conditions is under evaluation [122,123].

Chronic pain arises through a complex pathogenic process involving more components and developing into the pain continuum. Central sensitization, peripheral sensitization, and somatization are pathogenic processes of pain development in the pain continuum spanning components of the pain pathway and the pain mechanism, which is hardly understood without the presence of the cortical perception (Figure 5). The nociplastic mechanism of pain attempts to delineate pain without relevant cause or lesions of the somatosensory nervous system, such as altered perception of nociception. Chronic pain presented in fibromyalgia syndrome, chronic back pain, and complex regional pain syndrome is best understood in the framework of pain perception, including cognitive, emotional, and social components. Chronic pain experienced in psychiatric conditions, in particular, is not fully explainable in the view of the nociplastic pain mechanism. Pain sensation is developed through complex interactions with higher cortical centers governing mood, emotion, and cognition. 

Animal studies are one of the most important arenas for pain research. The endothelin receptor mediates the alteration of astrocyte functions, leading to the alleviation of neuropathic pain [124]. The dissociative anesthetics ketamine induces analgesic effects in models of acute pain and relieves thermal and mechanical allodynia in a chronic neuropathic pain model [125]. The involvement of the serotonergic neurotransmission in analgesic actions has been studied using neuropathic pain models in rats [126]. The gender difference in pain sensation and emotional domain has been reported using the transgenic mouse model of Alzheimer’s disease [127].

More and more emerging findings shed light on the relationship between psychiatric symptoms and networks of the brain centers in neuropsychiatric disorders [128,129,130]. Stimulus-evoked functional magnetic resonance imaging (fMRI), task-free fMRI, and perfusion MRI revealed that chronic pains arise from pre-existing vulnerabilities and sustained abnormal input [131]. Neuroimaging techniques, including fMRI and positron emission tomography, may open the gate to understanding underlying mechanisms in signaling to the third-order neurons to the cortex in chronic pain sensation [132,133,134]. Pain relief can be achieved through accompanying symptoms such as cognition, mood, and sleep by pharmacotherapy and/or psychotherapy [132,135,136,137]. Therefore, psychogenic components of pain play an essential role in understanding the pathomechanism of chronic pain unless the nociplastic pain mechanism can sufficiently elucidate the reciprocal interaction with third-order neurons in the pathogenesis of chronic pain.

## Figures and Tables

**Figure 1 biomedicines-09-00897-f001:**
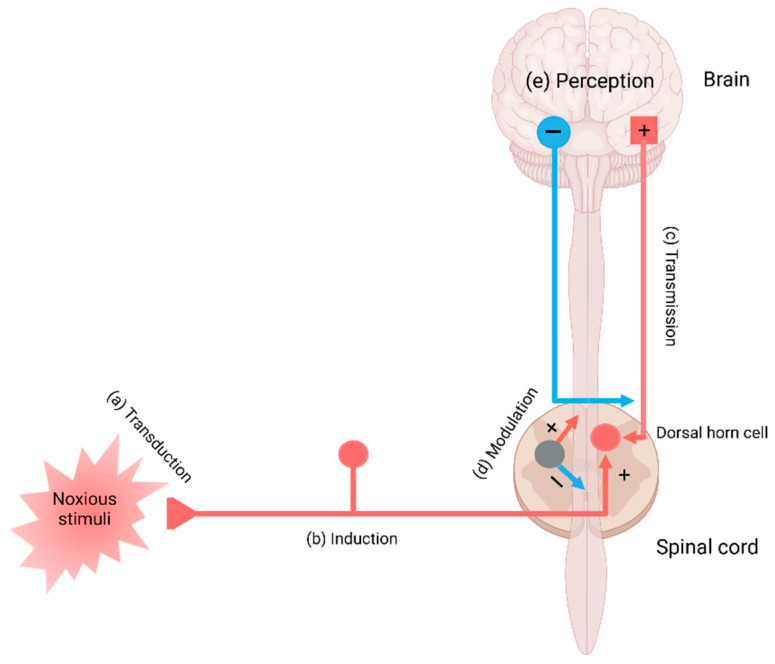
The main components in the pain pathway: (**a**) transduction, (**b**) induction, (**c**) transmission, (**d**) modulation, and (**e**) perception. Created with BioRender.com.

**Figure 2 biomedicines-09-00897-f002:**
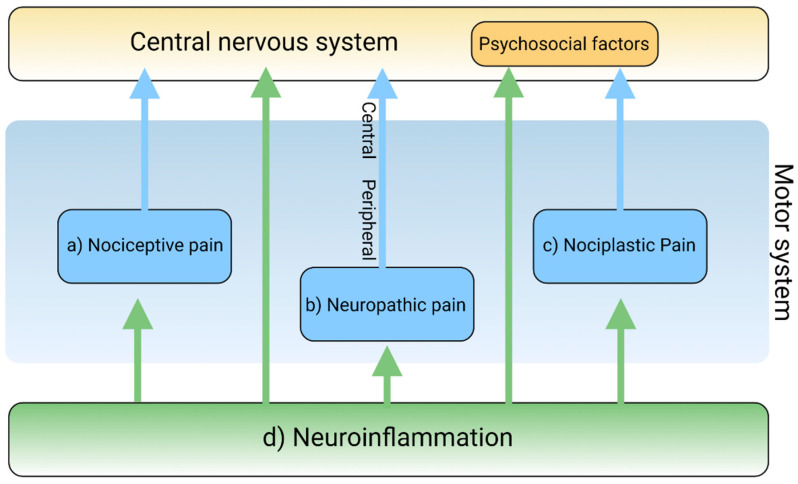
The main mechanisms of pain and the involvement of neuroinflammation. The mechanisms of pain are classified into (**a**) nociceptive, (**b**) neuropathic, and (**c**) nociplastic pain. Neuroinflammation (**d**) is involved in each pain mechanism. Created with BioRender.com.

**Figure 3 biomedicines-09-00897-f003:**
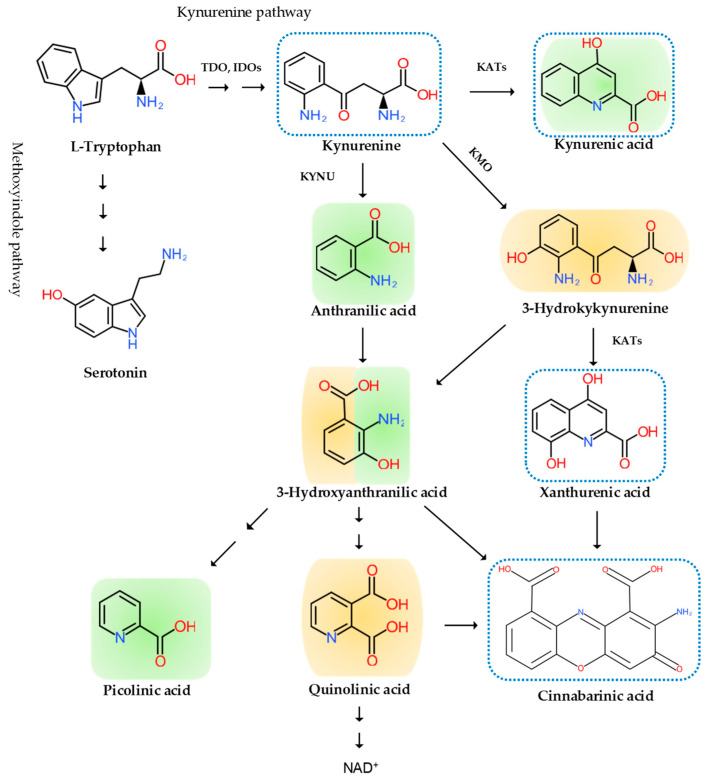
The tryptophan (TRP)–kynurenine (KYN) metabolic pathway and its metabolites. The pathway varies from the type of cells. Some enzyme is missing in some cells, not producing some metabolites. The TRP–KYN metabolic pathway synthesizes various metabolites, including oxidants (**orange color**), antioxidants (**green color**), and immunomodulators (**blue dotted line**). TDO: tryptophan 2,3-dioxygenase; IDO: indoleamine 2, 3-dioxygenase; KYNU: kynureninase; KMO: kynurenine-3-monooxygenase; KATs: kynurenine aminotransferases.

**Figure 4 biomedicines-09-00897-f004:**
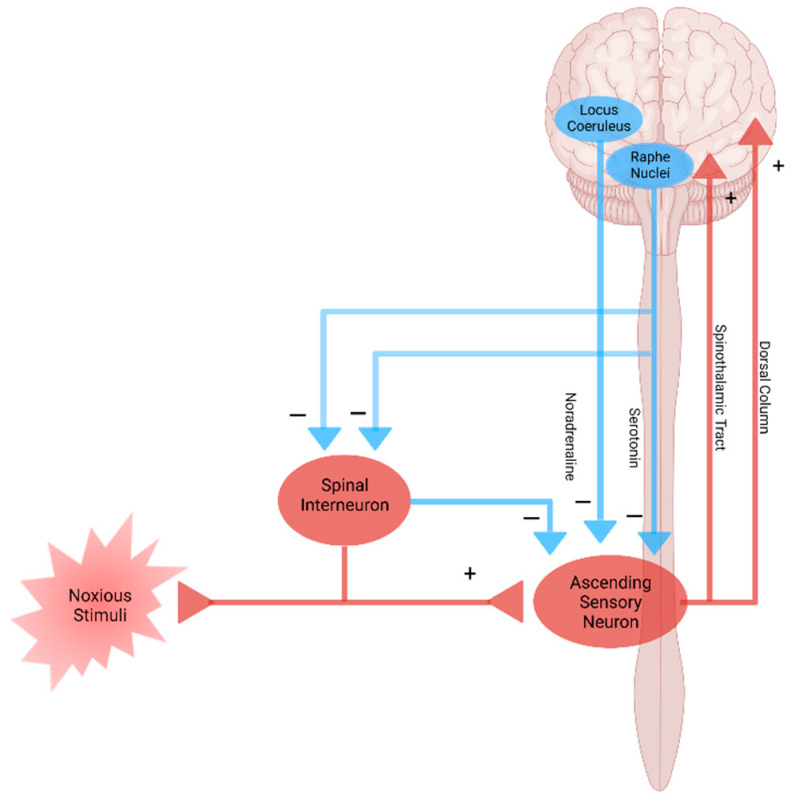
Pain pathway and pain modulation. Nociplastic pain is caused by the disturbance of central pain processing mechanisms, such as elevated excitability of ascending and descending pain facilitatory pathways and/or reduced inhibition of the descending anti-nociceptive pathway. Created with BioRender.com.

**Figure 5 biomedicines-09-00897-f005:**
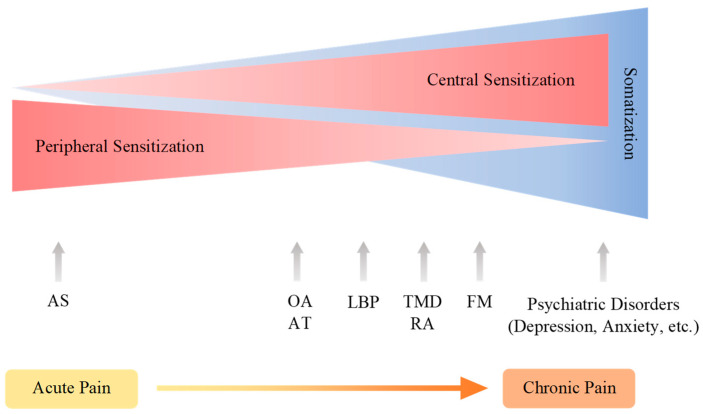
The continuum of pain sensitization and somatization. Chronic pain arises through a complex pathogenic process involving more components and developing into the pain continuum. Central sensitization, peripheral sensitization, and somatization are pathogenic processes of pain development spanning components of the pain pathway and the pain mechanism, which is hardly understood without the presence of the cortical perception. Acute pain may develop into chronic pain. AS: ankle sprain, OA: osteoarthritis, AT; Achilles tendinopathy; LBT: low back pain, TMD: temporomandibular joint disorder, FM: fibromyalgia.

**Table 1 biomedicines-09-00897-t001:** Pain pathway components, pain mechanisms, and representative diseases.

Pain Pathway Components	Pain Mechanisms	Diseases, Disorders, and Injuries
Transduction	Nociceptive pain	Ankle sprain and osteoarthritis
Conduction transmission	Neuropathic pain	Diabetic neuropathy, shingles, nutritional deficiencies, toxins, cancer, Guillain-Barre syndrome, amyloidosis, Fabry’s disease, and nerve trunk injuries
Modulation	Nociplastic pain	Fibromyalgia temporomandibular disorders, and nonspecific back pain
Perception	Psychogenic pain	Depression, anxiety, and cognitive impairment

## Data Availability

Not applicable.

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
