# Peer review of "Co-Players in Chronic Pain: Neuroinflammation and the Tryptophan-Kynurenine Metabolic Pathway"

_biomedicines, 2021, doi:10.3390/biomedicines9080897_

Round 1
Reviewer 1 Report
This well-organized review extensively presents the potential role of the brain area higher than the spinal cord in chronic pain. This manuscript indicated that in addition to the well-characterized nociceptive and neuropathic pain, impaired pain modulation could cause nociplastic pain, such as fibromyalgia and temporomandibular pain as well as modified pain perception might lead to psychogenic pain. Moreover, the authors have extensive reviewed the potential neurotransmitters/modulators involved in the nociceptive, neuropathic, nociplastic, and psychogenic pain. In particular, the serotonin-associated typtophan-kynurenine pathway has been discussed in detail.
Major comments:
This review could provide information about the development and maintenance of chronic pain caused by neural areas higher than the spinal cord level.
This manuscript is well-prepared and comprehensive, and references have been appropriately and adequately cited.
Minor comments:
Schematic diagrams using a circle to present the brain, brain stem (Pons varolli), and spinal cord that is somewhat confusing. The authors are encouraged to revised these diagrams using icons that resemble the shape of specific brain areas.
Author Response
Comments and Suggestions for Authors
This well-organized review extensively presents the potential role of the brain area higher than the spinal cord in chronic pain. This manuscript indicated that in addition to the well-characterized nociceptive and neuropathic pain, impaired pain modulation could cause nociplastic pain, such as fibromyalgia and temporomandibular pain as well as modified pain perception might lead to psychogenic pain. Moreover, the authors have extensive reviewed the potential neurotransmitters/modulators involved in the nociceptive, neuropathic, nociplastic, and psychogenic pain. In particular, the serotonin-associated typtophan-kynurenine pathway has been discussed in detail.
-Thank you for your summary. We are all grateful that the manuscript can convey main message.
Major comments:
This review could provide information about the development and maintenance of chronic pain caused by neural areas higher than the spinal cord level.
This manuscript is well-prepared and comprehensive, and references have been appropriately and adequately cited.
-We all appreciate your kind compliment.
Minor comments:
Schematic diagrams using a circle to present the brain, brain stem (Pons varolli), and spinal cord that is somewhat confusing. The authors are encouraged to revised these diagrams using icons that resemble the shape of specific brain areas.
-The figures are reversed to make them more visual. We hope the quality of the figures is improved.
Reviewer 2 Report
The review "Co-players in Chronic Pain: Neuroinflammation and the Tryptophan-Kynurenine Metabolic Pathway" brings together interesting aspects of pain classification. The frequent reminders of immune and neural system associations are, in my opinion, very important. The three categories of pain are well described, with examples and physiological background.
After reading this review, some questions seem to be left open:
- Is the tryptophan (TRP)-kynurenine (KYN) pathway the only major factor in pain systems? This topic plays an important role in this paper. Is it possible to explain the main impact of tis pathway? There are other small molecules involved in immune system/neural system interactions (small peptides, opioids, certain natural products)?
- There is an impact on pain classification (ICD etc.) in the introduction. It may be interesting to mention if there are any "measurements" available to detect/classify pain (or exclude certain reasons/types) - not to leave the reader with the idea that only the analysis of mediators is possible. (for example,
"Central Mechanisms of Pain Revealed Through Functional and Structural MRI" JOURNAL OF NEUROIMMUNE PHARMACOLOGY, 8(3), 518-534 2013
There are also some problems in the text:
Line 58: Hyperalgesia is a condition of abnormally increased sensitivity to
pain caused by injury to tissues or nerves, or by opioid medication for pain treatment.
This statement about opioids is important and may need some explanation, to bring the reason of hyperalgesia formation. Setting it at the same level as injury does not seem right.
Figure 2. The figure is difficult to study (small letters, limited differences). It will benefit from re-thinking the graphical concept.
Line 188: Ji et al - is there a reason to single out this reference?
Table 1. Column caption "Diseases" - is it the best caption if injuries are included?
Is the word "Singles" used intentionally in this table?
Line 238: "Lifetime and current prevalence of psychiatric disorders in patients with chronic peripheral pain were 39% and 20%, respectively" A reference to this stateent is needed - or put closely.
Line 352: "The levels of proinflammatory cytokines including IL-1RA, .....". IL-1RA is IL-1 receptor antagonist (please update the abbr table) - and calling it proinflammatory is not correct (without deep discussion on IL-1 regulation)?
Line 386: "Furthermore, chronic pain is associated with cognitive impairments such as attention, working memory, reasoning ability, and information processing [88]." This sentence has to be corrected.
Line 412: "representative diseases of pain are .." needs re-phrasing.
Line 424: There is no figure 5 in the text.
Figure 4 is not well connected/related to the text.
Reference 109 - please correct it as first names are left instead of names for some of the authors.
The extensive number of references added in Conclusion should be reconsidered.
Author Response
Comments and Suggestions for Authors
The review "Co-players in Chronic Pain: Neuroinflammation and the Tryptophan-Kynurenine Metabolic Pathway" brings together interesting aspects of pain classification. The frequent reminders of immune and neural system associations are, in my opinion, very important. The three categories of pain are well described, with examples and physiological background.
-Thank you for your critical reading. We all appreciate your work.
After reading this review, some questions seem to be left open:
Is the tryptophan (TRP)-kynurenine (KYN) pathway the only major factor in pain systems? This topic plays an important role in this paper. Is it possible to explain the main impact of tis pathway? There are other small molecules involved in immune system/neural system interactions (small peptides, opioids, certain natural products)?
- A new paragraph is added in the end of the Section 2 as follows: Stress hormone cortisol, strong immune activator lipopolysaccharide, proinflammatory cytokines, positive feedback loops, and diminished levels of antioxidant system enzyme superoxide dismutase and anti-inflammatory cytokines all lead to the potentiation of the TRP-KYN pathway. Furthermore, the action of the KYN enzymes and metabolites are complicated by the interactions with adjacent biosystems such as the oxidative stress complex, the antioxidant enzyme systems, the serotonin neurotransmission, the glutamate neurotransmission, the tetrahydrobiopterin pathway, the cannabinoid system, and the aryl hydrocarbon receptor (AHR) signaling.
There is an impact on pain classification (ICD etc.) in the introduction. It may be interesting to mention if there are any "measurements" available to detect/classify pain (or exclude certain reasons/types) - not to leave the reader with the idea that only the analysis of mediators is possible. (for example,
"Central Mechanisms of Pain Revealed Through Functional and Structural MRI" JOURNAL OF NEUROIMMUNE PHARMACOLOGY, 8(3), 518-534 2013
-The reference is cited, and the passage is added to Conclusion and Future Perspective as follows: Stimulus-evoked functional magnetic resonance imaging (fMRI), task-free fMRI, and perfusion MRI revealed that chronic pains arise from pre-existing vulnerabilities and sustained abnormal input.
There are also some problems in the text:
Line 58: Hyperalgesia is a condition of abnormally increased sensitivity to pain caused by injury to tissues or nerves, or by opioid medication for pain treatment. This statement about opioids is important and may need some explanation, to bring the reason of hyperalgesia formation. Setting it at the same level as injury does not seem right.
-The sentence is revised as follows: “The nociceptive sensation is also caused by exposure to opioids used for pain treatment, which paradoxically make more sensitive to certain stimuli.”
Figure 2. The figure is difficult to study (small letters, limited differences). It will benefit from re-thinking the graphical concept.
-The figure is revised accordingly. We hope that the new figure is more informativ.
Line 188: Ji et al - is there a reason to single out this reference?
-It is corrected accordingly.
Table 1. Column caption "Diseases" - is it the best caption if injuries are included?
-It is revised as follows: “Diseases, disorders, injuries”.
Is the word "Singles" used intentionally in this table?
-It is corrected to “Shingles”.
Line 238: "Lifetime and current prevalence of psychiatric disorders in patients with chronic peripheral pain were 39% and 20%, respectively" A reference to this stateent is needed - or put closely.
-The reference is added accordingly.
Line 352: "The levels of proinflammatory cytokines including IL-1RA, .....". IL-1RA is IL-1 receptor antagonist (please update the abbr table) - and calling it proinflammatory is not correct (without deep discussion on IL-1 regulation)?
-It was revised accordingly: “… IL-6 and IL-8 were observed to be higher, while anti-inflammatory cytokines IL-1RA was higher and Il-4 was lower…”
-IL-1RA is added in abbreviation table.
Line 386: "Furthermore, chronic pain is associated with cognitive impairments such as attention, working memory, reasoning ability, and information processing [88]." This sentence has to be corrected.
-The sentence is revised as follows: “… chronic pain is associated with the disturbance of cognitive functions such as attention…”.
Line 412: "representative diseases of pain are .." needs re-phrasing.
The passage is rephrased as follows: “… diseases associated with chronic pain …”.
Line 424: There is no figure 5 in the text.
-It is corrected.
Figure 4 is not well connected/related to the text.
-More description is added to the caption to explain the figure.
Reference 109 - please correct it as first names are left instead of names for some of the authors.
-The names of authors are double-checked.
The extensive number of references added in Conclusion should be reconsidered.
-The title of the section is changed to “Conclusion and Future Perspective” to cover more topics in addition to conclusion.